# META-TASKS: IMPROVING ROBUSTNESS IN FEW-SHOT CLASSIFICATION WITH UNSUPERVISED AND SEMI-SUPERVISED LEARNING

## ABSTRACT

Few-shot learning (FSL) is a challenging problem in machine learning due to the limited availability of labeled data. A major obstacle to FSL is the ability to generalize well on both novel tasks and training tasks. In this paper, we propose a new branch of unsupervised and semi-supervised regularization tasks to combat this problem. Our approach leverages both labeled and unlabeled data to improve the robustness and generalization performance of FSL models. Experimental results demonstrate the effectiveness of our proposed method by showing faster and better convergence, lower generalization, and standard deviation error both on novel tasks and training tasks, highlighting its potential for practical applications in FSL. Our proposed approach offers a promising solution to address the challenge of regularization in FSL, paving the way for future research in this area.

## 1 INTRODUCTION

Few-Shot Learning (FSL) is a promising method for dealing with the issue of limited data availability in computer vision tasks. FSL approaches use meta-learning techniques to learn how to learnWortsman et al. (2019) from a small number of instances, allowing them to generalize successfully to new tasks with limited training data. The main challenge in FSL is to develop efficient meta-learning algorithms that can generalize well to new tasks and acquire meaningful representations from a small number of samples. Recent developments in deep learning, notably in the field of meta-learning, have led to the development of powerful FSL techniques such as Matching NetworksVinyals et al. (2016), Relation NetworksSung et al. (2018), MAMLFinn et al. (2017), Prototypical NetworksSnell et al. (2017), Reptile Nichol & Schulman (2018). FSL enables models to learn about anomalies and recognize uncommon occurrences, which can be particularly useful in medical applications, such as locating uncommon diseases like COVID-19. By minimizing the quantity of data needed to train a model, FSL can drastically reduce the expenses involved with data gathering and annotation.Li et al. (2017); Setlur et al. (2020); Zhang et al. (2018).

Meta-learning has achieved impressive results in a wide range of domains, but thorough regularization is required to prevent overfitting and ensure good generalization performance both on novel samples from training classes and novel classes. Regularization is particularly difficult in meta-learning since these algorithms train exhaustively on a limited number of samples, which increases the chance of overfitting. Various regularization methods have been used in meta-learning, such as dropout, weight decay, and batch normalization, but their efficacy varies depending on the specific meta-learning algorithm and task at hand. Moreover, selecting the suitable regularization approach and associated hyperparameters can have a considerable impact on the meta-learning model's performance. As a result, developing effective regularization strategies that generalize well across different meta-learning algorithms and tasks remains a serious challenge.

The goal of FSL is to learn an adequate representation of the data such that new data can be classified accurately even with a few labeled samples. The first part of a few-shot learning method involves embedding the data into a high-dimensional feature space in a way that captures the important aspects of the data. The second part involves using the labeled examples to learn a classifier that can accurately predict the labels of new, unlabeled examples. In many few-shot learning methods, the embedding and classification stages are often combined into a single neural network architecture.

Although the two parts are closely related, they can still be viewed as separate components of a typical FSL method (Figure 2). The key contribution of this study is (1) the development of meta-tasks (Figure 1), a new sort of Meta-tasks that can regularize individual tasks or episodes. We argue that regularization can be considered a task in itself, and hence can be added as an additional loss function to the existing loss functions in a meta-learning setting. Furthermore, we propose that instead of traditional regularization approaches, unsupervised or semi-supervised machine learning tasks be used as regularization terms. For instance, by restricting the embedding vector to be reconstructable to the original image, an autoencoderKramer (1991) can be used as a regularizer for FSL. (2) Our research also presents a new regularization task for meta-learners called "meta-autoencoder". We show that using a meta-autoencoder leads to faster convergence with higher accuracy, lower generalization error, and standard deviation, highlighting its potential for practical applications in FSL.

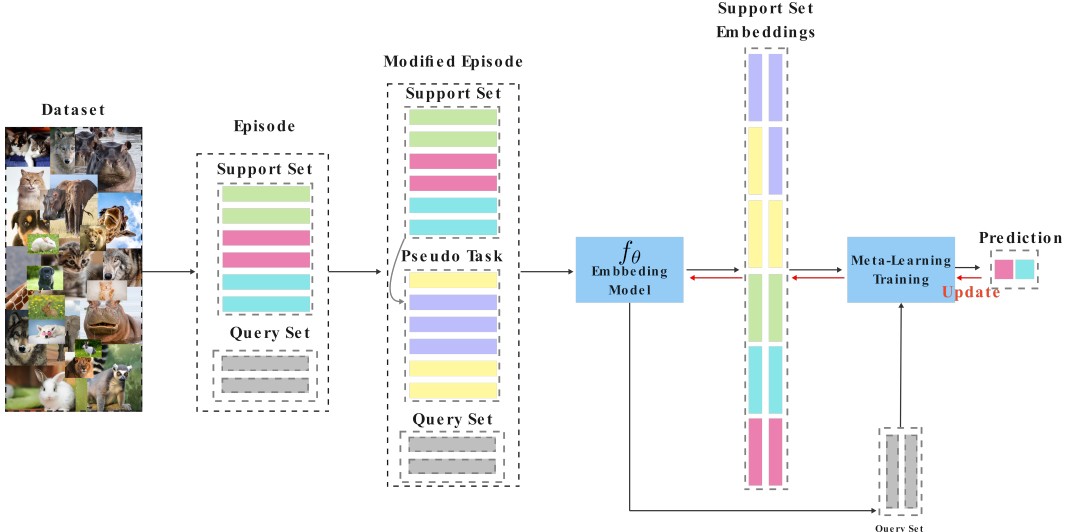

Figure 1: A High-level view of Meta-Task:
Each episode is made up of labeled examples (the support set) and unlabeled instances (the query set). The episode samples are then utilized to generate meta-task samples that are unique to that episode. The episode is subsequently processed by an embedding model, which maps the samples to a feature space and produces a compact representation of the episode. The meta-learning approach then uses this feature representation to learn and extract relevant information from the support set. Using the learned patterns and relationships from the support set, the obtained knowledge is then applied to predict or make inferences on the query set.

## 2 BACKGROUND/RELATED WORK

Meta-learning has been the subject of much research in recent years, and a variety of regularization methods have been proposed to improve the generalization capability of meta-learning models. Most conventional regularization methods from other areas of machine learning such as weight decay Krogh & Hertz (1991), dropout Gal & Ghahramani (2016), and incorporating noise. Achille & Soatto (2018); Alemi et al. (2016); Tishby & Zaslavsky (2015) are still applicable to meta-learning. However, there are some regularization techniques more specialized for meta-learning:

Explicit regularization methods impose explicit regularization terms on the meta-learning update, such as iMAML Rajeswaran et al. (2019), MR-MAML Yin et al. (2019), and works like Pan et al. (2021); Wang et al. (2023). These methods directly constrain the model optimization process.

Data augmentation regularization regulates the meta-training data augmentation, like making modifications to individual tasks through noise or mixup Yamaguchi et al. (2023); Shu et al. (2023). Task augmentation and interpolation generate new tasks by interpolating between or augmenting existing

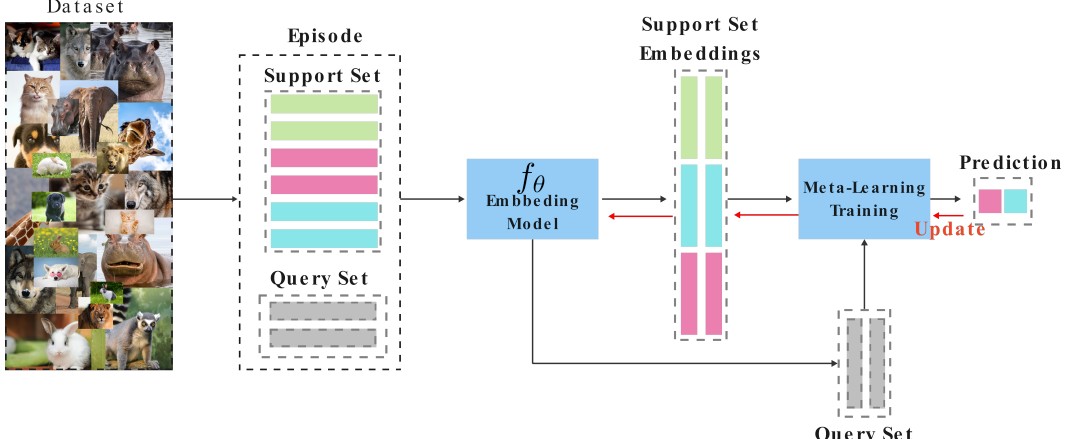

Figure 2: A High-level view of Meta-Learning:
Each episode is made up of labeled examples (the support set) and unlabeled instances (the query set). After that, the episode is fed into an embedding model, which maps the samples to a feature space, resulting in a compact representation of the episode. The meta-learning approach then uses this feature representation to learn and extract relevant information from the support set. Using the learned patterns and relationships from the support set, the obtained knowledge is then applied to predict or make inferences on the query set.

tasks Yao et al. (2021); Rajendran et al. (2020). The goal is to expand the task distribution for more robust generalization.

We hypothesize that by developing tasks as regularization methods, we can further enhance the generalization capability in meta-learning. Tasks have the potential to encode a diverse range of information about the world and the relationships between different concepts, enabling the model to learn generalizable patterns and ultimately improve its performance on new tasks. Our work aims to generalize the concepts of task augmentation and interpolation, transforming them into simple, explicit regularization terms that seamlessly integrate into existing meta-learning algorithms. While previous methods have introduced specialized techniques, we demonstrate that these can be generalized as meta-tasks, effectively acting as regularization mechanisms. Furthermore, our research reveals that even unrelated auxiliary tasks can serve as effective meta-tasks for regularization, extending their utility beyond traditional task interpolation.

Using tasks as regularization methods in meta-learning, as supported by previous research Yamaguchi et al. (2023); Shu et al. (2023) has the potential to improve generalization, reduce data requirements, enhance robustness, and expand applicability to various problems, findings that align with our experimental results.

## 3 PROBLEM STATEMENT

A meta-learning system has been proposed to deal with the problematic few-shot learning environment. The central concept is to learn how to adapt a base learner to a new task for which only a few labeled samples are available by using a large number of comparable few-shot tasks. Training numerous tasks are referred to as "meta", and learning to scale and modify the functions of Deep Neural Networks (DNN) weights for each task is based on transfer learning, meaning using the trained meta-learning model as a pre-trained model for each task Finn et al. (2017). Meta-learning often uses shallow neural networks (SNNs), which improves its effectiveness because it reduces intra-class variation Chen et al. (2019).

An $N$-way, $K$-shot classification consists of $N$ classes and $K$ examples of each class. They serve as the task's "support set" and aid the model in learning like a training set. The performance of this task is also assessed using additional examples of the same classes, which are referred to as a "query set". Samples from one task might not appear in others because one task can be entirely independent

of the others. The notion is that throughout training, the system continually encounters tasks with a structure similar to the final few-shot task but containing different classes. The meta-learning framework for few-shot learning involves selecting a random training task at each meta-learning step and updating model parameters based on the classification accuracy of the query set. The aim is to differentiate class labels in general rather than a specific subset of classes because the task at each time step is different. Few-shot performance is evaluated on a set of unseen classes, referred to as novel classes, that were not included in the training.

Prototypical networks Snell et al. (2017) are robust to data imbalance as they compute their mean embedding or prototype by averaging the embeddings of examples for each class. The prototypes are then classified based on their similarity with the query embedding, which is often calculated using cosine distance. While cosine distance typically outperforms Euclidean distance in prototypical networks, this may not always be the case, initialization can impact the performance of each distance metric Snell et al. (2017). Nonetheless, using a better weight initialization can lead to cosine distance typically outperforming Euclidean distance, as suggested by Chen et al. Chen et al. (2019). However, some restrictions remain for prototypical networks. One such constraint is the lack of generality, because neural networks may not perform well when attempting to identify images with diverse representations. While prototypical networks may perform well on the Omniglot dataset Lake et al. (2015), they may not yield trustworthy results when attempting to categorize distinct bird breeds due to representational differences. Another drawback of prototypical networks is that they only employ means to determine the center and neglect support set variance, which can compromise their ability to categorize images with noise. Nonetheless, prototypical networks remain popular due to their ability to produce outstanding results and represent a simpler inductive bias than contemporary few-shot learning methods, which is advantageous in environments with sparse data.

To address these limitations, we are exploring the addition of unsupervised tasks, such as decoder or adversarial modelsGoodfellow et al. (2020), to improve the model's understanding of new task-specific data. By incorporating these new techniques, prototypical networks may be able to overcome their constraints and achieve even higher levels of performance on few-shot learning tasks.

## 4 APPROACH

Let's suppose that the meta-dataset is split to train, validation, testing sets, and let's denote them by $D^{tr}$, $D^{val}$, $D^{te}$ where $D^{tr}$ and $D^{val}$ consist of tasks $T_1^{tr}, T_2^{tr}, \ldots, T_L^{tr}$ and our $D^{te}$ consist of tasks $T_1^{te}$, $T_2^{te}, \ldots, T_I^{te}$.

We introduce a new task, denoted as $T_{L1}^{tr}$, which only requires images from $D^{tr}$ and does not need labels, or labels can be created using available labels in $D^{tr}$. In each episode, we take $N$ tasks from $\{T_i^{tr}\}_{i=1}^{L}$ and use $T_{L1}^{tr}$ (and any other applicable $\{T_{Li}^{tr}\}$) as the regularizer for $\{T_i^{tr}\}_{i=1}^{L}$ or fine-tuned separately for each task.

To clarify this method, we can define a new task $T_{L1}^{tr}$, as generating an image in an autoencoder. In this task, the loss can be defined as the mean squared error (MSE) loss.

The goal of prototypical networks is to minimize the negative log probability (log-softmax loss). For a given query sample $x$, if $n$ is the correct label, then the loss function is defined as:

$$J_x(\theta) = -\log(p_\theta(y = n|x)) \tag{1}$$

where $p_\theta(y = n|x)$ is the probability of assigning the correct label $n$ to query sample $x$.

$$p_\theta(y = n|x) = \frac{\exp(-d(f_\theta(x), c_n)}{\sum_{k=1}^{N} \exp(-d(f_\theta(x), c_k))} \tag{2}$$

where $f_\theta$ is the embedding network parameterized by $\theta$, $c_n$ is the prototype of the $n$-th class, $d(a, b)$ is the Euclidean distance between vectors $a$ and $b$. The logarithm is used to increase the loss when the model fails to predict the correct class. Thus, we can rewrite the loss function and subsequently the update role as:

$$\min_\theta J_x(\theta) = d(f_\theta(x), c_n) + \log \sum_{k=1}^{N} \exp(-d(f_\theta(x), c_k)) \tag{3}$$

We add an additional term to the loss function to incorporate the regularization term into our method. This term is the loss function of the new task we introduced, which we denote as $T_{L1}^{\text{tr}}$. In our case, $T_{L1}^{\text{tr}}$ is the generation of an image using an autoencoder, and we use mean squared error as the loss function for this task. To be more specific, we add the autoencoder update role to the original update role of Prototypical Networks, resulting in the following loss function for a query sample $x$ with correct label $n$:

$$J_x(\theta) = \underbrace{d(f_\theta(x), c_n) + \log \sum_{k=1}^{N} \exp(-d(f_\theta(x), c_k))}_{\text{Meta-Learning Update}} + \overbrace{\lambda ||g_{\theta'}(f_\theta(x)) - x||}^{\text{AutoEncoder Update}} \tag{4}$$

where $g$ is the decoder network with parameters $\theta'$, and $\lambda$ is the learning rate for the autoencoder. This regularization term can be added to any other arbitrary method.

$$J_x(\theta) = -\log(p_\theta(y = n|x)) + \lambda ||g_{\theta'}(f_\theta(x)) - x|| \tag{5}$$

We calculate the sum of losses overall query samples in a task $T_i^{\text{tr}}$ to obtain the overall loss for that task:

$$J_{T_i^{\text{tr}}}(\theta) = \sum_{x \in T_i^{\text{tr}}} J_x(\theta) \tag{6}$$

During each episode, we compute the loss function for each task and sum them to get the overall loss functionFinn et al. (2017):

$$J(\theta) = \sum_{i=1}^{L} J_{T_i^{\text{tr}}}(\theta) \tag{7}$$

Now if we assume that we have a custom meta-task regularizer for each task by additive separability, we can rewrite this equation as:

$$J(\theta) = \sum_{i=1}^{L} \left( J_{T_i^{\text{tr}}}(\theta) + \lambda J_{T_{Li}^{\text{tr}}}(\theta) \right) \tag{8}$$

If the regularization term is separable (like the autoencoder update), we can simplify this to:

$$J(\theta) = \sum_{i=1}^{L} J_{T_i^{\text{tr}}}(\theta) + \lambda \sum_{i=1}^{L} J_{T_{Lr_i}^{\text{tr}}}(\theta) \tag{9}$$

Now, if want to generalize this equation to form where each task may have multiple or no meta-task attached to it. then again if these tasks are separable, We will have:

$$J(\theta) = \sum_{i=1}^{L} J_{T_i^{\text{tr}}}(\theta) + \lambda \sum_{r=1}^{R} J_{T_{Lr}^{\text{tr}}}(\theta) \tag{10}$$

Although we mostly discuss updates on prototypical networks, it is not strictly necessary to use our approach with them and it generalizes to any FSL method. Nonetheless, given recent developments in the field of few-shot learning, discussing updates on prototypical networks could be beneficial in highlighting the potential for continued progress in this area.

The logic behind incorporating meta-tasks is similar to that of meta-learning, as it aims to help a meta-learner better understand the commonalities between different tasks, learn to generalize more effectively and develop a more robust set of features that can be applied to new tasks. By training on a set of related meta-tasks, a meta-learner can leverage the knowledge gained from these tasks to learn more efficiently and adapt more rapidly to new situations.

Although the use of encoder-decoders is not new in meta-learning Rusu et al. (2018), we will use a novel autoencoder method to illustrate the effectiveness of this regularization, we have developed a method called Meta-Autoencoder. In this method, we use the MSE loss of the decoder as the regularization term for each episode. In other words, for each episode, we pass the images from the episode through the decoder to generate the regularization term for the update of the prototypical network. This regularization helps to improve the generalization performance of the model by

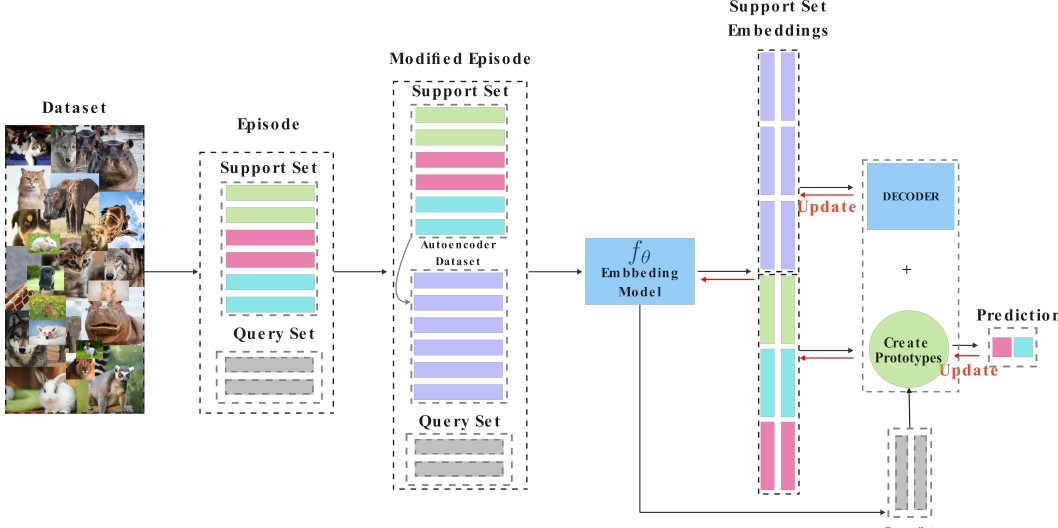

Figure 3: Detailed architecture of Meta-Autoencoder for a 3-Way 2-Shot FSL.
The training examples in the autoencoder task are identical copies of the support set samples but without their labels. For these examples, the embedding model generates feature vectors, which are then employed in two ways. They are first processed through a decoder to reconstitute the original input images, which aids the model in learning meaningful representations. Second, the feature vectors are used to generate prototypes for each support set class, which are then used to categorize query photos based on their similarity. The training procedure includes two backpropagation steps: one for updating model parameters based on the classification loss, which measures the match between prototypes and query images, and another for updating model parameters based on the decoder's reconstruction loss, which ensures accurate reconstruction of the original images.

encouraging the latent representations learned by the encoder to be more robust and informative. It is worth noting that the main distinction from Rusu et al. (2018) lies in our focus on regulating the training process, whereas their approach revolves around finding a better starting point using the episode.

Our findings demonstrate that our proposed method is effective in improving the training process via the incorporation of a decoder update as a regularization term. Specifically, we observed that introducing artificial tasks during training can lead to improved performance and standard deviation. These results suggest that our approach holds promise for enhancing the training of meta-learning methods in various applications.

## 5 RESULTS AND DISCUSSIONS

For our study, we used a ResNet50 architecture He et al. (2016) and implemented a ResNet-based autoencoder. This autoencoder was trained on bird 525 species dataset. The trained encoder was then utilized as the embedding model for our experiments. An NVIDIA Tesla P100 was used as a computing resource for model training. The experiments were conducted over $50,000$ episodes with a learning rate of $10^{-4}$ and a query size of $15$. Additionally, the autoencoder was trained with a separate learning rate of $10^{-6}$. The complete source code and results for this paper can be accessed at the following GitHub repository: HIDDEN

To assess the effectiveness of different approaches, we conducted a comparative study between a Prototypical Network and a meta-autoencoder Prototypical Network. Initially, we tuned the hyperparameters of both models for the 5-shot 5-way task on the miniImageNet dataset Ravi & Larochelle (2017). These adjusted hyperparameters were then used for all subsequent training runs. It is worth emphasizing that assessing few-shot learning methods is difficult due to a lack of appropriate evaluation metrics. Standard measurements of accuracy and precision, which are designed for non-repetitive samples, may not be appropriate for few-shot learning scenarios. For our evaluation, we

relied on the available metrics for comparing meta-learning methods as they are currently the primary tools for comparison. Additionally, we also examined the performance of the two models using traditional evaluation metrics, assuming each sample to be unique, which can be found in the supplementary materials. The results of our study are summarized in the following tables 1 2 3.

**Tiered ImageNet**

**Train**

| | 5-Shot 5-Way | | | | 5-Shot 1-Way | | | |
| --- | --- | --- | --- | --- | --- | --- | --- | --- |
| | Prototypical Network | | Meta-AutoEncoder | | Prototypical Network | | Meta-AutoEncoder | |
| # Episodes | Accuracy | Loss | Accuracy | Loss | Accuracy | Loss | Accuracy | Loss |
| 10000 | 61.00% | 1.2871±0.0002 | 58.70% | 1.0339±0.0000 | 41.30% | 1.4699±0.0002 | 39.30% | 1.4194±0.0000 |
| 20000 | 64.70% | 1.2506±0.0002 | 68.10% | 0.8196±0.0000 | 44.80% | 1.4382±0.0002 | 46.30% | 1.2950±0.0000 |
| 30000 | 65.70% | 1.2406±0.0002 | 71.30% | 0.7395±0.0000 | 45.40% | 1.4329±0.0002 | 49.80% | 1.2235±0.0000 |
| 40000 | 66.00% | 1.2371±0.0002 | 73.50% | 0.6855±0.0000 | 45.90% | 1.4295±0.0002 | 51.90% | 1.1790±0.0000 |
| 50000 | 66.00% | 1.2387±0.0002 | 74.60% | 0.6579±0.0000 | 46.10% | 1.4268±0.0002 | 53.80% | 1.1376±0.0000 |

**Validation**

| | 5-Shot 5-Way | | | | 5-Shot 1-Way | | | |
| --- | --- | --- | --- | --- | --- | --- | --- | --- |
| | Prototypical Network | | Meta-AutoEncoder | | Prototypical Network | | Meta-AutoEncoder | |
| # Episodes | Accuracy | Loss | Accuracy | Loss | Accuracy | Loss | Accuracy | Loss |
| 10000 | 58.90% | 1.3089±0.0002 | 62.60% | 0.9583±0.0000 | 40.40% | 1.4848±0.0002 | 42.40% | 1.3606±0.0000 |
| 20000 | 60.70% | 1.2905±0.0002 | 65.50% | 0.8850±0.0000 | 41.80% | 1.4681±0.0002 | 44.60% | 1.3107±0.0000 |
| 30000 | 60.80% | 1.2878±0.0002 | 67.60% | 0.8390±0.0000 | 42.20% | 1.4636±0.0002 | 46.10% | 1.2780±0.0000 |
| 40000 | 61.50% | 1.2818±0.0002 | 68.90% | 0.8198±0.0000 | 42.70% | 1.4576±0.0002 | 47.40% | 1.2528±0.0000 |
| 50000 | 61.30% | 1.2854±0.0002 | 68.00% | 0.8206±0.0000 | 41.90% | 1.4625±0.0002 | 49.70% | 1.2265±0.0000 |

**Test**

| | 5-Shot 5-Way | | | | 5-Shot 1-Way | | | |
| --- | --- | --- | --- | --- | --- | --- | --- | --- |
| | Prototypical Network | | Meta-AutoEncoder | | Prototypical Network | | Meta-AutoEncoder | |
| # Episodes | Accuracy | Loss | Accuracy | Loss | Accuracy | Loss | Accuracy | Loss |
| 10000 | 62.60% | 1.2712±0.0002 | 64.30% | 0.9061±0.0000 | 42.90% | 1.4568±0.0002 | 42.70% | 1.3961±0.0000 |
| 20000 | 64.50% | 1.2522±0.0002 | 67.90% | 0.8174±0.0000 | 45.60% | 1.4345±0.0002 | 45.90% | 1.3094±0.0000 |
| 30000 | 64.70% | 1.2488±0.0002 | 69.30% | 0.7859±0.0000 | 46.00% | 1.4271±0.0002 | 47.30% | 1.2810±0.0000 |
| 40000 | 65.40% | 1.2427±0.0002 | 70.20% | 0.7643±0.0000 | 46.60% | 1.4228±0.0002 | 49.30% | 1.2290±0.0000 |
| 50000 | 65.30% | 1.2453±0.0002 | 71.30% | 0.7437±0.0000 | 46.70% | 1.4226±0.0002 | 50.00% | 1.2319±0.0000 |

Table 1: The table displays the training accuracy of the prototypical network and meta-autoencoder on tiered-ImageNet. The results of our experiments clearly show that our method maintains consistent accuracy and loss compared to the prototypical network.

The experiments demonstrate that our proposed method outperforms the prototypical network, as evidenced by the lower generalization error and standard deviation on both the training and test sets. Additionally, we explored different hyperparameters such as the number of episodes, ways, shots, and learning rate, and found that the results remained consistent. However, we also observed that increasing the number of episodes and learning rate could lead to overfitting, particularly when working with a larger learning rate. It is important to consider the risk of overfitting during training. Furthermore, it is worth noting that simply adding more meta-tasks does not necessarily result in better regularization, as meta-tasks are not inherently prone to task overfitting Pan et al. (2021).

## 6 CONCLUSION

In this paper, our primary objective was to address the regularization problem in Few-Shot Learning (FSL) methods. To achieve this, we introduced a novel approach called Meta-Task, which serves as an additional regularization task that can be easily incorporated into any FSL method. This Meta-Task is constructed using semi-supervised or unsupervised techniques such as autoencoders. we focused on the N-way K-shot image classification problem and extensively investigated the impact of Meta-Tasks on improving model performance. By incorporating Meta-Tasks, we observed a significant reduction in the noise present in the input data, leading to faster convergence and lower generalization error. Our findings highlight the effectiveness of using autoencoders to enhance accuracy while simultaneously reducing loss. By thoroughly exploring the concept of Meta-Tasks and their relationship with the meta-learning problem, we have provided a clear understanding of how this approach can effectively address the regularization problem. Meta-tasks offer a convenient and efficient way to improve the performance of FSL methods by leveraging additional auxiliary tasks. Our results demonstrate the benefits of incorporating Meta-Tasks, emphasizing their potential for enhancing model accuracy and reducing overfitting.

**Mini-ImageNet**
**Train**

| | 5-Shot 5-Way | | | | | 5-Shot 1-Way | | | |
| | Prototypical Network | | Meta-AutoEncoder | | | Prototypical Network | | Meta-AutoEncoder | |
| # Episodes | Accuracy | Loss | Accuracy | Loss | Accuracy | Loss | Accuracy | Loss |
|---|---|---|---|---|---|---|---|---|
| 10000 | 73.20% | 0.6973±0.0004 | 66.40% | 0.8635±0.0001 | 51.20% | 1.2019±0.0005 | 45.60% | 1.3094±0.0000 |
| 20000 | 77.20% | 0.5971±0.0004 | 75.80% | 0.6362±0.0000 | 54.70% | 1.1227±0.0005 | 54.00% | 1.1423±0.0000 |
| 30000 | 77.60% | 0.5873±0.0004 | 79.30% | 0.5476±0.0000 | 56.30% | 1.0853±0.0005 | 57.60% | 1.0592±0.0001 |
| 40000 | 78.00% | 0.5784±0.0004 | 81.70% | 0.4836±0.0000 | 56.30% | 1.0816±0.0005 | 60.40% | 0.9958±0.0001 |
| 50000 | 78.00% | 0.5747±0.0004 | 83.60% | 0.4345±0.0000 | 55.90% | 1.0921±0.0005 | 62.30% | 0.9513±0.0001 |

**Validation**

| | 5-Shot 5-Way | | | | | 5-Shot 1-Way | | | |
| | Prototypical Network | | Meta-AutoEncoder | | | Prototypical Network | | Meta-AutoEncoder | |
| # Episodes | Accuracy | Loss | Accuracy | Loss | Accuracy | Loss | Accuracy | Loss |
|---|---|---|---|---|---|---|---|---|
| 10000 | 63.10% | 1.0029±0.0004 | 64.60% | 0.9166±0.0000 | 42.10% | 1.3870±0.0004 | 44.80% | 1.3719±0.0000 |
| 20000 | 64.00% | 0.9762±0.0003 | 63.80% | 0.9047±0.0000 | 44.00% | 1.3305±0.0005 | 44.90% | 1.3460±0.0000 |
| 30000 | 65.10% | 0.9559±0.0003 | 66.60% | 0.9002±0.0000 | 45.00% | 1.3028±0.0004 | 47.80% | 1.3020±0.0001 |
| 40000 | 64.70% | 0.9439±0.0003 | 64.10% | 0.9429±0.0000 | 44.20% | 1.3163±0.0004 | 51.70% | 1.2375±0.0000 |
| 50000 | 65.30% | 0.9439±0.0003 | 64.90% | 0.9113±0.0000 | 44.50% | 1.3069±0.0004 | 51.60% | 1.2625±0.0000 |

**Test**

| | 5-Shot 5-Way | | | | | 5-Shot 1-Way | | | |
| | Prototypical Network | | Meta-AutoEncoder | | | Prototypical Network | | Meta-AutoEncoder | |
| # Episodes | Accuracy | Loss | Accuracy | Loss | Accuracy | Loss | Accuracy | Loss |
|---|---|---|---|---|---|---|---|---|
| 10000 | 63.60% | 0.9921±0.0003 | 62.80% | 0.8928±0.0000 | 41.90% | 1.3889±0.0004 | 42.00% | 1.3766±0.0000 |
| 20000 | 64.40% | 0.9713±0.0003 | 67.30% | 0.8296±0.0000 | 43.20% | 1.3449±0.0005 | 44.60% | 1.3597±0.0000 |
| 30000 | 65.30% | 0.9618±0.0003 | 66.10% | 0.8646±0.0000 | 45.00% | 1.3013±0.0004 | 45.40% | 1.3162±0.0000 |
| 40000 | 64.60% | 0.9510±0.0003 | 65.60% | 0.8583±0.0000 | 44.10% | 1.3229±0.0004 | 46.60% | 1.2753±0.0000 |
| 50000 | 65.20% | 0.9447±0.0003 | 65.80% | 0.8789±0.0000 | 44.60% | 1.3088±0.0005 | 47.90% | 1.2457±0.0000 |

Table 2: The table displays the training accuracy of the prototypical network and meta-autoencoder on tiered-ImageNet. When the losses are compared, the meta-autoencoder regularization outperforms the Prototypical Network alone in terms of training regularization. This addition reduces overall loss, indicating improved generalization performance. The autoencoder regularization term assists the model in learning more robust and informative latent representations through the encoder, resulting in better training regularization.

**FC100**
**Train**

| | 5-Shot 5-Way | | | | | 5-Shot 1-Way | | | |
| | Prototypical Network | | Meta-AutoEncoder | | | Prototypical Network | | Meta-AutoEncoder | |
| # Episodes | Accuracy | Loss | Accuracy | Loss | Accuracy | Loss | Accuracy | Loss |
|---|---|---|---|---|---|---|---|---|
| 10000 | 82.90% | 0.4462±0.0004 | 76.50% | 0.6114±0.0001 | 64.80% | 0.8841±0.0005 | 56.70% | 1.0698±0.0001 |
| 20000 | 86.30% | 0.3589±0.0003 | 84.80% | 0.3976±0.0000 | 68.70% | 0.7887±0.0005 | 66.80% | 0.8373±0.0001 |
| 30000 | 86.80% | 0.3417±0.0003 | 87.40% | 0.3291±0.0000 | 70.40% | 0.7464±0.0005 | 70.80% | 0.7385±0.0001 |
| 40000 | 87.20% | 0.3338±0.0003 | 89.20% | 0.2838±0.0000 | 70.30% | 0.7418±0.0005 | 73.30% | 0.6794±0.0001 |
| 50000 | 87.20% | 0.3336±0.0003 | 90.30% | 0.2540±0.0000 | 70.50% | 0.7447±0.0005 | 75.30% | 0.6303±0.0001 |

**Validation**

| | 5-Shot 5-Way | | | | | 5-Shot 1-Way | | | |
| | Prototypical Network | | Meta-AutoEncoder | | | Prototypical Network | | Meta-AutoEncoder | |
| # Episodes | Accuracy | Loss | Accuracy | Loss | Accuracy | Loss | Accuracy | Loss |
|---|---|---|---|---|---|---|---|---|
| 10000 | 47.20% | 1.3308±0.0004 | 49.80% | 1.2624±0.0000 | 30.50% | 1.7034±0.0003 | 31.10% | 1.6558±0.0000 |
| 20000 | 49.40% | 1.2897±0.0003 | 47.40% | 1.3148±0.0000 | 31.30% | 1.7150±0.0004 | 31.40% | 1.6751±0.0000 |
| 30000 | 49.10% | 1.2954±0.0004 | 48.50% | 1.3068±0.0000 | 30.30% | 1.7311±0.0003 | 30.50% | 1.7099±0.0000 |
| 40000 | 48.80% | 1.3078±0.0004 | 47.50% | 1.3024±0.0000 | 29.60% | 1.7471±0.0004 | 30.30% | 1.7077±0.0000 |
| 50000 | 49.30% | 1.2945±0.0003 | 48.50% | 1.3148±0.0000 | 29.50% | 1.7476±0.0004 | 31.30% | 1.7162±0.0000 |

**Test**

| | 5-Shot 5-Way | | | | | 5-Shot 1-Way | | | |
| | Prototypical Network | | Meta-AutoEncoder | | | Prototypical Network | | Meta-AutoEncoder | |
| # Episodes | Accuracy | Loss | Accuracy | Loss | Accuracy | Loss | Accuracy | Loss |
|---|---|---|---|---|---|---|---|---|
| 10000 | 47.20% | 1.3346±0.0004 | 49.10% | 1.2150±0.0000 | 30.90% | 1.6981±0.0003 | 38.50% | 1.5198±0.0000 |
| 20000 | 49.60% | 1.2937±0.0004 | 51.10% | 1.2090±0.0000 | 31.10% | 1.7164±0.0004 | 38.60% | 1.4865±0.0000 |
| 30000 | 48.80% | 1.2978±0.0004 | 50.90% | 1.1973±0.0000 | 30.30% | 1.7299±0.0004 | 39.60% | 1.5105±0.0000 |
| 40000 | 49.10% | 1.3037±0.0003 | 51.00% | 1.2762±0.0000 | 29.70% | 1.7430±0.0004 | 36.60% | 1.5620±0.0000 |
| 50000 | 48.70% | 1.3006±0.0004 | 49.20% | 1.2720±0.0000 | 29.30% | 1.7554±0.0004 | 39.00% | 1.5076±0.0000 |

Table 3: The table presents the training accuracy of both the prototypical network and meta-autoencoder on FC100 dataset. The results indicate that our method achieves faster convergence compared to the prototypical network. This is especially evident when observing the accuracy of the prototypical network over the next 10,000 episodes, where the meta-autoencoder consistently outperforms it.

we also introduced a novel Meta-Task called meta-autoencoder. The meta-autoencoder leverages the power of autoencoders within the meta-learning framework to enhance the learning process. By incorporating the meta-autoencoder task, we aim to improve the model's ability to extract meaningful features and representations from the input data. This novel approach offers a promising direction for advancing meta-learning algorithms and opens up new opportunities for improving performance in various domains. We recognized the importance of fine-tuning the learning rate for optimal performance and plan to explore this further in future work.

While meta-tasks are effective in addressing the issue of overfitting, it is important to note that they are not immune to overfitting themselves. It is crucial to find the right balance when incorporating meta-tasks into the meta-learning process. Adding too many meta-tasks can lead to task overfitting, where the model becomes overly specialized to the specific tasks in the meta-training set and fails to generalize well to new tasks. Therefore, careful consideration should be given to the selection and number of meta-tasks to ensure optimal performance and generalization ability of the meta-learning algorithm.

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

## A  APPENDIX

More information on the experiments can be found here.

| Prototypical Network | | | | |
|---|---|---|---|---|
| | *precision* | *recall* | f1-score | support |
| 0 | 29.12% | 28.49% | 28.80% | 22635 |
| 1 | 54.58% | 50.13% | 52.26% | 64455 |
| 2 | 26.88% | 29.89% | 28.31% | 32565 |
| 3 | 49.94% | 42.34% | 45.83% | 65115 |
| 4 | 36.33% | 33.04% | 34.60% | 23910 |
| 5 | 56.25% | 62.92% | 59.40% | 31275 |
| 6 | 53.73% | 41.41% | 46.77% | 39420 |
| 7 | 44.38% | 46.71% | 45.51% | 38310 |
| 8 | 26.97% | 30.00% | 28.40% | 31665 |
| 9 | 70.61% | 86.22% | 77.64% | 23130 |
| 10 | 53.81% | 49.29% | 51.45% | 54825 |
| 11 | 27.54% | 43.78% | 33.81% | 39690 |
| 12 | 34.26% | 34.13% | 34.19% | 46980 |
| 13 | 37.02% | 33.19% | 35.00% | 62745 |
| 14 | 30.11% | 19.93% | 23.99% | 32160 |
| 15 | 65.74% | 76.98% | 70.91% | 16020 |
| 16 | 56.42% | 51.69% | 53.96% | 24045 |
| 17 | *55.85%* | *80.23%* | 65.86% | 23205 |
| 18 | 46.94% | 38.72% | 42.44% | 39870 |
| 19 | 47.49% | 52.11% | 49.69% | 37980 |
| accuracy | 44.48% | | | |
| macro avg | 45.20% | 46.56% | 45.44% | 750000 |
| weighted avg | 44.75% | 44.48% | 44.23% | 750000 |
| meta-learning accuracy | 42.90% | | | |
| | | | | |

Table 4: 5-way 1-shot classification report

| Meta-autoencoder | | | | |
|---|---|---|---|---|
| | *precision* | *recall* | **f1-score** | **support** |
| 0 | 41.58% | 37.27% | 39.31% | 64200 |
| 1 | 49.30% | 58.49% | 53.51% | 47295 |
| 2 | 50.43% | 50.11% | 50.27% | 39165 |
| 3 | 49.96% | 29.91% | 37.42% | 16200 |
| 4 | 56.85% | 38.64% | 46.01% | 70305 |
| 5 | 62.96% | 66.98% | 64.91% | 24285 |
| 6 | 39.47% | 44.93% | 42.02% | 23100 |
| 7 | 44.45% | 46.86% | 45.62% | 39465 |
| *8* | *48.39%* | *39.82%* | 43.69% | 22965 |
| 9 | 50.20% | 55.14% | 52.55% | 54960 |
| 10 | 43.57% | 37.51% | 40.31% | 39480 |
| 11 | 27.81% | 22.48% | 24.87% | 24030 |
| 12 | 36.03% | 25.30% | 29.73% | 47070 |
| 13 | 35.88% | 36.83% | 36.35% | 47415 |
| 14 | 35.86% | 50.05% | 41.79% | 15885 |
| 15 | 59.57% | 85.49% | 70.21% | 38685 |
| 16 | 51.83% | 84.35% | 64.21% | 39405 |
| *17* | *57.98%* | *61.81%* | 59.84% | 47970 |
| 18 | 20.93% | 14.95% | 17.44% | 32655 |
| 19 | 54.04% | 54.30% | 54.17% | 15465 |
| accuracy | 47.28% | | | |
| macro avg | 45.86% | 47.06% | 45.71% | 750000 |
| weighted avg | 46.47% | 47.28% | 46.14% | 750000 |
| meta-learning accuracy | 47.30% | | | |
| | | | | |

Table 5: 5-way 1-shot classification report

| Hyper parameters | | |
|---|---|---|
| *epoch* | *5* | *5* |
| train_num_episode | 10000 | 10000 |
| test_num_episode | 10000 | 10000 |
| train_way | 5 | 5 |
| train_shot | 5 | 1 |
| train_query | 15 | 15 |
| test_way | 5 | 5 |
| test_shot | 5 | 1 |
| test_query | 15 | 15 |
| *optimizer* | *Adam* | *Adam* |
| autoencoder_lr | 1.00E-06 | 1.00E-06 |
| lr | 1.00E-04 | 1.00E-04 |

