# OpenReview forum: "Meta-Tasks: Improving Robustness in Few-Shot Classification with Unsupervised and Semi-Supervised Learning"
_ICLR.cc/2024/Conference — ICLR 2024 Conference Withdrawn Submission_

### Official Review · Reviewer_EG2x · 2023-10-30

**Soundness:** 2 fair
**Presentation:** 2 fair
**Contribution:** 1 poor
**Rating:** 1
**Confidence:** 5

**Summary:**

This paper introduces the unsupervised autoencoder as a regularization into meta-learning methods. The experiments show that the method can bring some improvement.

**Strengths:**

- The method is simple and easy to understand.

**Weaknesses:**

- Unsupervised learning/pretraining has long proven to be very effective for few-shot learning [1,2,3]. The method proposed in this paper is simply adding an additional self-supervised reconstrution loss into meta-learning, which is trivial.
- The motivation for using the reconstruction loss is not clear. The proposed method can be used for common classification tasks as well, so what makes it special to work for few-shot learning?
- The writing of the paper needs significant improvement. The paper is like a draft or a project in the university. For example, Figure 1,2,3 are very similar and should be put into one figure. Descriptions of the method can be much shorter. The definition of regularization needs to be clarified more. The logic of the paper needs to be rearranged.
- The experiments only show that the additional loss can improve meta-learning's performance. However, no other methods are compared, making it hard to put the method into the context of the literature. Also, ablation study is missing.

[1] Boosting Few-Shot Classification with View-Learnable Contrastive Learning. ICME 2021.

[2] Boosting Few-Shot Visual Learning with Self-Supervision. ICCV 2019.

[3] A Closer Look at Few-shot Classification Again. ICML 2023.

**Questions:**

I suggest the authors see more recent papers on few-shot learning literature, especially [1,2], to get a better sense of what's going on in the field currently, and what are the main problems that the community cares about.

[1] A Closer Look at Few-shot Classification Again. ICML 2023.
[2] Pushing the Limits of Simple Pipelines for Few-Shot Learning: External Data and Fine-Tuning Make a Difference. CVPR 2022.

---

### Official Review · Reviewer_Ka51 · 2023-10-30

**Soundness:** 2 fair
**Presentation:** 1 poor
**Contribution:** 1 poor
**Rating:** 3
**Confidence:** 4

**Summary:**

This paper introduces an Auto-Encoder into the classic few-shot classification method (ProtoNet) to incorporate a self-regression loss as a regularization term, enhancing the classification capability of few-shot learning.

**Strengths:**

The proposed method is simple to implement and easy to follow.

**Weaknesses:**

1. The content in the first three sections is somewhat redundant and does not effectively highlight the contributions of this work.

2. This paper is not the first one to explore the effect of self-supervision task in few-shot learning. Please refer to [1][2].

3. The motivation is not clear. The authors fail to demonstrate why self-supervision loss can be a good regularization.

4. The experiments are not convinced. 1) This paper appears to propose a general viewpoint: self-supervision can enhance performance in few-shot tasks, but it does not explore various self-supervision methods and few-shot methods for a more comprehensive comparison and evaluation. 2) Lack of essential ablative experiments: It is possible that the Auto-encoder itself is a powerful few-shot learner, which means that the performance of the Auto-encoder may already surpass that of ProtoNet + Auto-encoder. However, the paper itself does not confirm this.

5. The template of ICLR 2023 shouldn't be used.




**Reference**

[1] Spyros Gidaris, Andrei Bursuc, Nikos Komodakis, Patrick Pérez, Matthieu Cord.Boosting Few-Shot Visual Learning with Self-Supervision. In ICCV 2019.

[2] Jong-Chyi Su, Subhransu Maji, Bharath Hariharan. When Does Self-supervision Improve Few-shot Learning? In ECCV 2020.

**Questions:**

Please refer to the section of **Weakness**.

---

### Official Review · Reviewer_S9AD · 2023-10-30

**Soundness:** 2 fair
**Presentation:** 3 good
**Contribution:** 2 fair
**Rating:** 3
**Confidence:** 4

**Summary:**

This paper proposes a new branch of unsupervised and semi supervised regularization tasks for few-shot learning. Their method utilizes labeled and unlabeled data to improve the robustness and generalization performance of few-shot learning models. And limited experiment shows the improvement brought by the proposed method.

**Strengths:**

The writing is good, the proposed method is well-illustrated.

**Weaknesses:**

1. Lack of novelty. Introducing the meta-task regularization is not novel in few-shot learning. And the proposed Meta-Autoencoder is also not novel.
2. Lack of verification. This paper only deploys the proposed method on prototypical network and compares the results on three datasets. No experimental results can show the performance consistency on other FSL method. And no compared regularization method to prove the effectiveness.
3. Need improvement on way of showing experimental results. The Table 1-3 can be better illustrated with figure.

**Questions:**

As mentioned before.

---

### Official Review · Reviewer_ooCT · 2023-10-31

**Soundness:** 1 poor
**Presentation:** 2 fair
**Contribution:** 2 fair
**Rating:** 3
**Confidence:** 4

**Summary:**

The paper explores the problem of improving few-shot image classification accuracy by introducing an auxiliary task during episodic training to improve performance. This meta-task is to use the shared feature space through an autoencoder to introduce an additional loss term that aims to reconstruct the query examples. Authors claim that this additional loss term acts as a regularizer, thereby producing more robust features for few-shot learning. Experiments on mini- and tiered-ImageNet and FC100 demonstrate that when the proposed framework is adapted inside Prototypical Networks, performance in improved.

**Strengths:**

- With some exceptions noted below, the language of the paper is clear and relatively straightforward to follow
- Use of an image reconstruction task through an autoencoder is reasonable and shown to improve performance empirically

**Weaknesses:**

- The submission overclaims the generalizability of their method while the results are only demonstrated when the meta-autoencoder is adapted inside Prototypical Networks; Prototypical Networks, although very popular, are only one few-shot learning framework and the authors should adjust their claims to reflect this, or provide experimental evidence for using their method inside other few-shot learning architectures.
- The use of auxiliary tasks when training few-shot learning algorithms is not novel and the proposed framework to use query feature vectors through an autoencoder is relatively straightforward. As a result, the technical contributions of the paper are incremental.
- Experimental accuracies reported do not contain confidence intervals or any statistical measure of significance. As a result, the statistical significance of the results is not appropriately established.
- Overall, the submission as it stands, makes very generalized claims about the usefulness of the proposed meta-task framework across many architectures and tasks but only demonstrates them in comparison to Prototypical Networks. Discussion of and comparison to other baselines [1-6] would be important in establishing the empirical validity of their method.

[1] Matching Networks for One Shot Learning
[2] Model-Agnostic Meta-Learning for Fast Adaptation of Deep Networks
[3] Learning to Compare: Relation Network for Few-Shot Learning
[4] TADAM: Task dependent adaptive metric for improved few-shot learning
[5] Improved Few-Shot Visual Classification
[6] Meta-Learning with Latent Embedding Optimization

**Questions:**

- During episodic training, does the decoder in the autoencoder also receive gradient updates or is it kept fixed?
- Please also address the weaknesses noted above. Although, as it stands, I would recommend against the acceptance of the paper, should the authors adequately address the limitations noted above, I would be more than happy to improve my current rating.

---

### Meta-Review · Area_Chair_7ZAz · 2023-12-06

**Metareview:**

The paper proposes to  leverage both labeled and unlabeled data to improve the robustness and generalization performance of FSL models. As the reviewers pointed out, using auxiliary tasks to enhance the training of few-shot learning models has been intensively studied and the proposed method does not bring much novel insight to advance the state of the art. As a result, the overall technical contribution appears to be limited. The evaluation results are not very convincing, either.

**Justification For Why Not Higher Score:**

Lack of novelty and less convincing evaluation results.

**Justification For Why Not Lower Score:**

N/A

---

### Decision · Program_Chairs · 2024-01-16

Reject